# Solubility of Methane in Ionic Liquids for Gas Removal Processes Using a Single Multilayer Perceptron Model

Claudio A. Faúndez [1], Elías N. Fierro [1,*] and Ariana S. Muñoz [2]

[1] Departamento de Física, Universidad de Concepción, Casilla 160-C, Concepción 3349001, Chile; claudiofaundez@udec.cl
[2] Facultad de Ingeniería, Universidad Autónoma de Chile, 5 Poniente 1670, Talca 3480094, Chile; ariana.munoz@uautonoma.cl
* Correspondence: elfierro@udec.cl

**Abstract:** In this work, four hundred and forty experimental solubility data points of 14 systems composed of methane and ionic liquids are considered to train a multilayer perceptron model. The main objective is to propose a simple procedure for the prediction of methane solubility in ionic liquids. Eight machine learning algorithms are tested to determine the appropriate model, and architectures composed of one input layer, two hidden layers, and one output layer are analyzed. The input variables of an artificial neural network are the experimental temperature (T) and pressure (P), the critical properties of temperature (Tc) and pressure (Pc), and the acentric ($\omega$) and compressibility (Zc) factors. The findings show that a (4,4,4,1) architecture with the combination of T-P-Tc-Pc variables results in a simple 45-parameter model with an absolute prediction deviation of less than 12%.

**Keywords:** $CO_2$; solubility; methane; ionic liquids; artificial neural network; multilayer perceptron; algorithm learning





## 1. Introduction

Knowledge of the solubility of gases in ionic liquids (ILs) is essential for the design and development of separation processes and the evaluation of their environmental impact [1–3]. The present need for more and better data and correlations to estimate the properties of different types of mixtures, including ILs, has inspired our interest in studying, analyzing, and proposing sound methods to determine the solubility of gases in ionic liquids, which are of interest in environmental industrial applications. Knowledge of the mixing behavior of gases and ILs is highly relevant for their potential use in several chemical processes [4].

Gases such as $CO_2$, $CH_4$, $NH_3$, $H_2S$, $SO_2$, and $N_2O$, among others, are frequently present as contaminants in process streams while producing, processing, and refining petroleum fractions [1,2,5–9] The selective and efficient removal of contaminants such as those mentioned above and other impurities from synthesis gases is of special importance to make them suitable for downstream processes. Thus, the solubility of the gases in low-volatility or nonvolatile solvents is highly relevant for many technological applications [10,11]. Due to their extraordinary solvent qualities, ILs seem to be good alternative solvents to be explored.

Accurate data, correlations, and models of the solubility of gases in ILs are required for the design and operation of removal processes. However, there are no experimental data in the literature and no general methods for correlating and predicting solubility in a general form [12,13].

Thus, the development of accurate models for correlating and predicting the phase behavior of such systems is of current interest. Certainly, there are no predicted data that can replace good experimental values, but sometimes such good experimental data are not readily available, especially for systems containing ILs.

Some of the reasons that have motivated interest in ILs are (i) their very low vapor pressure, (ii) their highly polar nature, (iii) the selective solubility of some ILs for certain components in fluid mixtures, and (iv) the fact that their physical and chemical properties depend directly on the particular combination of anions and cations that form the ILs [14,15]. During the last ten years, research on different aspects and applications of ILs has notably increased. Studies including the synthesis of new ILs, evaluation of their environmental, impact and toxicity, and technical and economic analyses, as well as experimental and theoretical research to obtain physical and physicochemical properties, appear every day in the scientific literature [16]. The phase behavior of mixtures involving ILs is also receiving increasing attention, and several studies on aspects such as liquid-liquid equilibrium, liquid-gas equilibrium, the thermodynamic consistency of data, solubility parameters, experimental difficulties, miscibility, and the effects of impurities have appeared in the literature [17–25].

In recent years, our research group has explored different approaches to correlate and predict several properties of pure ILs and mixtures containing ILs. Modeling and predicting mixture properties, including ILs, represent new and different challenges that are now possible to explore due to the availability of the necessary critical properties to apply current phase equilibrium and liquid-liquid methods. We have proposed new models to correlate and predict the solubility of gases in ILs [23–28]. The natural next step in this line of research is the study of the properties of mixtures involving ILs, such as the solubility of methane in the ILs that are analyzed in this work.

Methane ($CH_4$) is one of the main components of natural gas and plays a major role in supplementing the energy required in each society. While high-purity $CH_4$ is sold to the commodity natural gas market or converted to other chemicals, such as methanol or carbon black, medium-purity methane is used for generating electricity or significant heat for various processes [3]. Methane is one of the main greenhouse gases. Therefore, whether as a priceless energy resource or a strong destructive gas, $CH_4$ absorption in chemical processes is highly relevant [3,29]. The solubility of $CH_4$ in low-volatility or nonvolatile solvents is thus highly relevant for many technological applications. However, experimental data on the solubility of $CH_4$ in ILs available in the literature are scarce [8].

Until now, there have been no general methods to correlate, estimate, and predict the solubility of $CH_4$ in ILs. There are, however, several studies showing vapor–liquid equilibrium data for gases and ILs mixtures, data that are used in this study. Additionally, chemical engineers have been able to use powerful simulation programs to estimate how a given process behaves under certain operating conditions. This type of software allows us to correlate and predict the phase behavior of gases in ILs, but at present, it does not include the necessary basic data (pure component properties, solubility parameters, and Henry's law constants) that allow us to simulate processes involving ILs. The lack of this information will certainly delay further development of industrial processes and the transfer to the industry of technologies that use IL mixtures.

Some studies related to the vapor–liquid equilibria (VLE) of $CH_4$ and IL mixtures that will be useful in the development of new proposals have been presented in the literature. Loreno et al. (2019) studied the solubility of $CH_4$ in four imidazolium-based ILs in the temperature range from 293 K to 363 K. The experimental results were correlated using the group contribution–simplified perturbed chain–statistical associating fluid theory (GC-sPC-SAFT) [30]. Hamedi et al. (2020) correlated the solubility of $CH_4$ in 19 different ILs in the temperature range from 293 K to 449 K. The $CH_4$ solubility values were predicted using two statistical equations of state models [3]. Kurnia et al. (2020b) determined the solubility of methane in four alkylpyridinium-based ILs in the temperature range from 298 K to 343 K and pressures up to 4 MPa [8]. Huang (2022) reported an experimental study and a simulation study on the capture and separation of methane by alkali metal ILs. The range of temperatures considered in the study was from 303 to 323 K, and the pressures were from 500 to 3000 kPa [31].

In recent years, artificial neural networks (ANNs) have become a new tool for the study of thermodynamic properties. These networks, inspired by how biological neurons function, have proven to be as efficient as traditional thermodynamic models. Safamirzaei and Modarress (2012) correlated and predicted the solubility of $CH_4$ and six other gases in [bmim][$BF_4$], using the molecular weight, acentric factor, and experimental temperature and pressure as input variables [32]. Behera et al. (2015) predicted a performance parameter, namely methane percentage (%), using a multilayer perceptron model. The landfill gas (LFG) extraction rate ($m^3$/h) and landfill leachate/food waste leachate (FWL) ratio parameters, which were obtained from a field-scale investigation, were input into the network [33]. Nair et al. (2016) investigated the performance of a laboratory-scale anaerobic bioreactor to determine the $CH_4$ content in biogas yield from the digestion of an organic fraction of municipal solid waste by applying an ANN model, using free forward backpropagation [34]. Dashti et al. (2018) determined the solubility values of natural gas composed of $CH_4$ and carbon dioxide for 11 different ILs, using hybrid ANN models, namely coupled simulated annealing–least square support vector machine (CSA-LSSVM) and particle swarm optimization–adaptive neuro-fuzzy inference system model (PSO-ANFIS) [35].

Conventional thermodynamic methods require a considerable number of adjustable parameters. For example, the Peng–Robinson cubic equation of state with Kwak–Mansoori mixing rule requires three adjustable parameters for each temperature. On the other hand, works reported in the literature on gas solubilities in ILs using ANN show similar results to those obtained with conventional methods, but with the characteristic of being simple models with a reduced number of parameters [7,25].

In the present work, 440 experimental vapor–liquid equilibrium data points of 14 systems composed of $CH_4$ and ILs are employed to train a multilayer perceptron model. To ensure efficient training of the neural network without sacrificing the model's generalization capability, the temperature and pressure ranges considered in this work are 293.15–449.30 K and 0.400–16.105 MPa, respectively. An ANN is trained with eight learning algorithms, and four combinations of inputs are studied during the training process. In this study, only systems whose critical properties have been reported by Valderrama et al. (2015) were selected [36]. To select the optimal architecture, in addition to the usual statistical criteria, simple models with respect to the number of parameters are considered. As a result, a simple model is found that allows us to predict the solubility of systems composed of $CH_4$ and ILs with high accuracy.

## 2. Multilayer Perceptron

A multilayer perceptron (MLP) is a structure composed of processing units named neurons. Each neuron is assigned a weight (w) and a bias (b), which are parameters of the network. Neurons are organized into three layers: input layer, hidden layer, and output layer. Figure 1 shows the diagram of the MLP used in this work. The data are processed by means of connections between all the neurons in one layer and all the neurons in the next layer. In this work, the output of the *l*-neuron in the hidden layer ($k + 1$) of a network with *M* layers is given by Equation (1) [37,38].

$$a_l^{k+1} = f^{k+1}\left(\sum_{j=1}^{N_k} w_{lj}^{k+1} a_l^k + b_l^{k+1}\right); k = 1, 2, 3. \ldots .(M - 1) \tag{1}$$

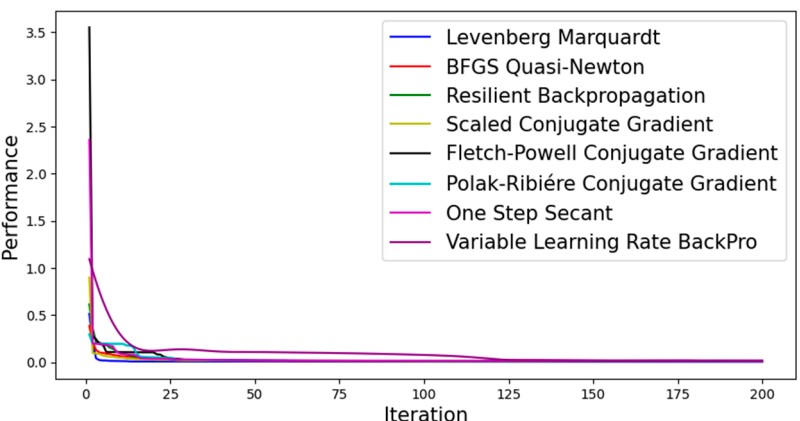

**Figure 1.** Convergence of the 8 algorithms used in the training of the (4,2,2,1) architecture and T, P, Tc, and Pc input combination.

For complicated, high-dimensional, nonlinear datasets, an activation function is employed. In this work, the activation function for the hidden layers is the *tansig* function given by Equation (2), and that for the output layer is the linear *purelin* function given by Equation (3):

$$f(x)_{tansing} = \frac{e^x - e^{-x}}{e^x + e^{-x}} \tag{2}$$

$$f(x)_{purelin} = n \tag{3}$$

The weights (w) and the bias (b) must be adjusted to minimize the error of the output (Yi). In this work, the objective function is the mean square error (MSE), which is given by Equation (4),

$$MSE = \frac{1}{N}\sum_{i=1}^{n}(X_i - Y_i)^2 \tag{4}$$

The individual absolute deviation, average absolute deviation, and average relative deviation of each solubility calculated with respect to the experimental data are determined using Equations (5)–(7), respectively.

$$|\Delta x\%| = \frac{100}{N}\sum_{i=1}^{N}\left(\frac{x_i^{cal} - x_i^{exp}}{x_i^{exp}}\right) \tag{5}$$

$$|\Delta \overline{x_1}\%| = \frac{100}{N}\sum_{i=1}^{N}\left|\frac{x_i^{cal} - x_i^{exp}}{x_i^{exp}}\right| \tag{6}$$

$$\Delta \overline{x_1}\% = \frac{100}{N}\sum_{i=1}^{N}\left(\frac{x_i^{cal} - x_i^{exp}}{x_i^{exp}}\right) \tag{7}$$

## 3. Results and Discussion

In the present work, we study 14 binary systems composed of $CH_4$ in different ILs. Table 1 shows the temperature, pressure, and solubility ranges for each system considered in this work, indicating the literature sources from which the experimental data were selected. The temperatures vary between 293.15 K and 449.12 K, the pressure varies between 0.400 MPa and 16.105 MPa, and the solubility varies between 0.001 and 0.496. The study includes 440 experimental data points (P–T-x data).

**Table 1.** All the data considered in this work.

| System | N | T(K) | P(MPa) | x1 | Reference |
|---|---|---|---|---|---|
| [C4mim][Tf$_2$N] | 8 | 300.31–314.31 | 1.510–16.105 | 0.030–0.225 | |
| | 5 | 332.58–342.31 | 1.618–10.503 | 0.030–0.163 | |
| | 4 | 352.00–352.08 | 3.237–10.982 | 0.056–0.163 | |
| | 5 | 371.33–371.38 | 1.736–11.352 | 0.030–0.163 | [39] |
| | 5 | 390.69–400.47 | 1.836–11.652 | 0.030–0.163 | |
| | 5 | 410.09–410.22 | 3.583–8.440 | 0.056–0.122 | |
| | 5 | 429.56–429.80 | 3.659–11.978 | 0.056–0.122 | |
| | 5 | 448.96–449.12 | 1.938–12.054 | 0.030–0.163 | |
| [C4py][BF$_4$] | 6 | 298.15 | 1.670–3.910 | 0.012–0.026 | |
| | 5 | 313.15 | 1.770–3.910 | 0.011–0.023 | [8] |
| | 5 | 328.15 | 2.140–4.120 | 0.012–0.022 | |
| | 4 | 343.15 | 2.370–3.890 | 0.012–0.020 | |
| [C4py][Tf$_2$N] | 7 | 298.15 | 0.900–4.120 | 0.011–0.049 | |
| | 6 | 313.15 | 1.360–4.030 | 0.014–0.047 | |
| | 7 | 328.15 | 1.030–4.150 | 0.011–0.042 | |
| | 6 | 343.15 | 1.310–3.990 | 0.012–0.036 | |
| [C6py][Tf$_2$N] | 7 | 298.15 | 0.700–3.990 | 0.013–0.074 | |
| | 7 | 313.15 | 0.720–3.940 | 0.012–0.066 | |
| | 7 | 328.15 | 0.930–3.990 | 0.014–0.060 | |
| | 6 | 343.15 | 1.270–3.920 | 0.019–0.054 | |
| [C2mim][dep] | 5 | 303.17–303.44 | 1.685–8.310 | 0.020–0.076 | |
| | 5 | 313.13–313.24 | 1.755–8.565 | 0.020–0.076 | |
| | 5 | 323.06–323.25 | 1.820–8.800 | 0.020–0.076 | |
| | 5 | 333.02–333.25 | 1.880–9.020 | 0.020–0.076 | [40] |
| | 5 | 343.02–343.25 | 1.930–9.176 | 0.020–0.076 | |
| | 5 | 353.05–353.25 | 1.975–9.316 | 0.020–0.076 | |
| | 5 | 362.92–363.29 | 2.025–9.441 | 0.020–0.076 | |
| [C2mim][FAP] | 3 | 293.30–293.58 | 2.076–5.831 | 0.052–0.129 | |
| | 4 | 303.29–303.57 | 2.151–7.728 | 0.052–0.155 | |
| | 4 | 313.42–313.54 | 2.185–7.951 | 0.052–0.155 | |
| | 4 | 323.36–323.52 | 2.234–8.123 | 0.052–0.155 | |
| | 4 | 333.27–333.47 | 2.299–8.321 | 0.052–0.155 | [29] |
| | 4 | 343.32–343.45 | 2.368–8.484 | 0.052–0.155 | |
| | 4 | 353.24–353.44 | 2.392–8.583 | 0.052–0.155 | |
| | 4 | 363.13–363.42 | 2.421–8.692 | 0.052–0.155 | |
| [C6mim][NO$_3$] | 5 | 293.15 | 0.874–2.580 | 0.020–0.087 | |
| | 5 | 303.15 | 0.905–2.680 | 0.021–0.089 | |
| | 5 | 313.15 | 0.935–2.778 | 0.022–0.091 | |
| | 5 | 323.15 | 0.966–2.876 | 0.022–0.093 | [41] |
| | 5 | 333.15 | 0.996–2.972 | 0.023–0.095 | |
| | 5 | 343.15 | 1.025–3.055 | 0.024–0.099 | |
| [C6mim][Tf$_2$N] | 8 | 298.15 | 0.400–0.999 | 0.012–0.028 | |
| | 7 | 313.15 | 0.501–0.998 | 0.012–0.027 | [42] |
| | 8 | 333.15 | 0.400–1.000 | 0.010–0.024 | |
| [C6mpy][Tf$_2$N] | 8 | 298.15 | 0.400–0.999 | 0.012–0.028 | |
| | 7 | 313.15 | 0.501–0.998 | 0.012–0.027 | |
| | 8 | 333.15 | 0.400–1.000 | 0.010–0.024 | |
| [tes][Tf$_2$N] | 5 | 303.10–303.43 | 1.246–7.039 | 0.024–0.111 | |
| | 5 | 312.87–313.25 | 1.301–7.314 | 0.024–0.111 | |
| | 5 | 322.85–323.25 | 1.351–7.534 | 0.024–0.111 | |
| | 5 | 333.11–333.28 | 1.391–7.749 | 0.024–0.111 | [40] |
| | 5 | 342.96–343.34 | 1.426–7.925 | 0.024–0.111 | |
| | 5 | 353.16–353.36 | 1.467–8.090 | 0.024–0.111 | |
| | 5 | 362.94–363.46 | 1.507–8.230 | 0.024–0.111 | |
| [thtdp][dca] | 7 | 302.13–303.38 | 1.428–9.638 | 0.079–0.343 | |
| | 7 | 312.14–313.39 | 1.503–10.049 | 0.079–0.343 | |
| | 7 | 322.25–323.47 | 1.576–10.433 | 0.079–0.343 | |
| | 7 | 332.29–333.52 | 1.651–10.759 | 0.079–0.343 | |
| | 7 | 342.31–343.53 | 1.697–11.059 | 0.079–0.343 | |
| | 7 | 352.36–353.46 | 1.752–11.329 | 0.079–0.343 | |
| | 7 | 362.34–363.48 | 1.792–11.569 | 0.079–0.343 | |

**Table 1.** *Cont.*

| System | N | T(K) | P(MPa) | x1 | Reference |
|--------|---|------|--------|-----|-----------|
| [thtdp][phos] | 6 | 302.00–303.25 | 1.015–9.708 | 0.107–0.496 | |
| | 6 | 311.98–313.25 | 1.066–10.173 | 0.107–0.496 | |
| | 6 | 322.05–323.19 | 1.131–10.628 | 0.107–0.496 | |
| | 6 | 332.02–333.22 | 1.171–11.023 | 0.107–0.496 | |
| | 6 | 342.05–343.25 | 1.216–11.404 | 0.107–0.496 | |
| | 6 | 352.11–353.26 | 1.261–11.734 | 0.107–0.496 | |
| | 6 | 362.15–363.27 | 1.301–12.049 | 0.107–0.496 | |
| [toa][Tf$_2$N] | 5 | 302.96–303.55 | 1.725–6.067 | 0.076–0.290 | |
| | 5 | 312.94–313.28 | 1.815–6.332 | 0.076–0.290 | |
| | 5 | 322.92–323.33 | 1.905–6.588 | 0.076–0.290 | |
| | 5 | 332.96–333.53 | 1.336–6.848 | 0.076–0.290 | |
| | 5 | 343.21–343.65 | 1.386–7.103 | 0.076–0.290 | |
| | 5 | 353.22–353.75 | 1.436–7.328 | 0.076–0.290 | |
| | 5 | 363.25–363.72 | 1.481–7.543 | 0.076–0.290 | |
| TMGL | 8 | 308.00 | 2.560–9.660 | 0.012–0.043 | |
| | 7 | 318.00 | 3.510–9.990 | 0.013–0.039 | [43] |
| | 7 | 328.00 | 3.690–10.340 | 0.010–0.031 | |

The method for predicting solubility is a multilayer perceptron (MLP) model composed of one input layer, two hidden layers, and one output layer. In the optimization of the ANN, the following stages are considered: the study of different learning algorithms, optimization of each hidden layer, and analysis of different input combinations. For the learning process of the MLP, the original available data are divided into three sets: 396 data points for training, 22 data points for testing, and 22 data points for prediction (each datasets can be found in the Supplementary Materials). To minimize overfitting in the prediction process, the datasets for the training, testing, and prediction stages were randomly selected.

In this study, the parameters statistic, average absolute deviation ($|\Delta x1\%|$), and maximum average absolute deviation ($|\Delta x1\%|_{max}$) are used as criteria for the selection of the best model, ensuring that the artificial neural network does not predict individual solubilities that are negative or greater than 1. Another criterion is to select ANN architectures with a reduced number of parameters.

*Learning Process*

In an MLP model, the most appropriate variables for solubility prediction have not been previously determined. However, different input combinations have been studied for solubility prediction [7,24,25,44–46]. For the choice of the algorithm, a simple architecture model is used with four input variables: the experimental temperature (T) and pressure (P); and the critical temperature (Tc) and pressure (Pc).

Table 2 shows the values of all critical properties used in this work. The ANN toolbox available in MATLAB 2023a is used to build an MLP with an (l,m,n,1) architecture, with one input layer (l = 4 and 5), two hidden layers (n = 2, 3, . . ., 10 and m = 2, 3, . . ., 10), and one output layer [47]. Table 3 shows the code used.

**Table 2.** Critical properties, acentric factors, and compressibility factors of all the substances used in this study [36].

| System | IUPAC Name | Tc | Pc | Zc | ω |
|--------|-----------|-----|-----|-----|-----|
| [C4mim][Tf$_2$N] | 1-Butyl-3-methylimidazolium bis(trifluoromethylsulfonyl)imide | 1258.9 | 27.64 | 0.2592 | 0.3370 |
| [C4py][BF$_4$] | 1-Butylpyridinium tetrafluoroborato | 597.6 | 20.33 | 0.2652 | 0.8207 |
| [C4py][Tf$_2$N] | 1-Butylpyridinium bis(trifluorometanosulfonyl)imide | 1229.1 | 27.71 | 0.2666 | 0.2505 |
| [C6py][Tf$_2$N] | 1-Hexylpyridinium bis(trifluorometanosulfonyl)imide | 1252.3 | 23.93 | 0.2522 | 0.3383 |
| [C2mim][dep] | 1-ethyl-3-methylimidazolium diethylphosphate | 877.2 | 21.47 | 0.2349 | 0.7219 |
| [C2mim][FAP] | 1-ethyl-3-methylimidazolium tris(perfluoroethyl)trifluorophosphate | 740.6 | 10.05 | 0.1944 | 1.3993 |

**Table 2.** *Cont.*

| System | IUPAC Name | Tc | Pc | Zc | ω |
|---|---|---|---|---|---|
| [C6mim][NO$_3$] | 1-Hexyl-3-methylimidazolium nitrate | 991.8 | 23.16 | 0.2135 | 0.7242 |
| [C6mim][Tf$_2$N] | 1-Hexyl-3-methylimidazolium bis(trifluoromethylsulfonyl)imide | 1293.3 | 23.89 | 0.2454 | 0.3874 |
| [hmpy][Tf$_2$N] | 1-Hexyl-1-methylpyrrolidinium bis(trifluorometanosulfonyl)imide | 1265.2 | 22.25 | 0.2439 | 0.4060 |
| [tes][Tf$_2$N] | triethylsulfonium bis(trifluoromethylsulfonyl)imide | 1189.9 | 21.90 | 0.2317 | 0.1603 |
| [thtdp][dca] | trihexyltetradecylphosphonium dicyanamide | 1505.8 | 7.65 | 0.1388 | 1.0319 |
| [thtdp][phos] | trihexyltetradecylphosphonium bis(2,4,4-trimethylpentyl)phosphinate | 1819.5 | 5.51 | 0.1157 | 0.0924 |
| [toa][Tf$_2$N] | methyltrioctylammonium bis(trifluoromethylsulfonyl)imide | 1347.6 | 10.64 | 0.1988 | 1.6063 |
| TMGL | 1,1,3,3-tetramethylguanidium lactate | 816.9 | 27.18 | 0.2557 | 1.1188 |

**Table 3.** The MATLAB code used in this work.

```
1% TRAINING SECTION%
2% Reading independent variables for training
3p = xlsread('variables_X1_training');p = p';
4%Reading the dependent variable for training;
5t = xlsread('X1_for_training');t = t';
6% Normalization of all data (values between −1 and y +1)
7[pn,minp,maxp,tn,mint,maxt] = premnmx(p,t);
8% Definition of ANN:(topology, activation functions, training algorithm)
9net = newff(minmax(pn),[6,6,1],{'tansig','tansig','purelin'},'trainlm');
10% Definition of frequency of visualization of errors during training
11net.trainParam.show = 10;
12% Definition of number of maximum iterations and global error between iterations
13net.trainParam.epochs = 1000; net.trainParam.goal = 1 × 10⁻⁴;
14% Network starts: reference random weights and gains
15w1 = net.IW{1,1}; w2 = net.LW{2,1}; w3 = net.LW{3,2};
16b1 = net.b{1}; b2 = net.b{2}; b3 = net.b{3};
17% First iteration with reference values and correlation coefficient
18before_training = sim(net,pn);
19corrbefore_training= corrcoef(before_training,tn);
20% Training process and results
21[net,tr] = train(net,pn,tn); after_training = sim(net,pn);
22% Back-Normalization of results, from values between −1 y and +1 to real values
23after_training = postmnmx(after_training,mint,maxt); after_training = after_training';
24Res = sim(net,pn);
25% Saving results, correlated solubility in an Excel file
26dmwrite('X1_correlated.xls',after_training,char(9));
27save w
28%TESTING SECTION
29%Reading weight and other characteristics of the trained ANN saved in the file W
30load w
31% Reading of Excel file with new independent variables to predict
32pnew = xlsread('variables_sol_ prediction'); pnew = pnew';
33% Normalization of all variables (values between −1 y and +1)
34pnewn = tramnmx(pnew,minp,maxp);
35% Testing the ANN obtaining the properties for the variables provided
36anewn = sim(net,pnewn);
37% Transformation of the normalized exits (between −1 y and +1) determined by the ANN to real values
38anew = postmnmx(anewn,mint,maxt); anew = anew';
39% Saving the testing properties in an Excel file
40dlmwrite('solub_ testing.xls',anew,char(9));
41%PREDICTION SECTION
42%Reading weight and other characteristics of the trained ANN saved in the file W
43load w
44% Reading of Excel file with new independent variables to predict
45pnew = xlsread('variables_sol_ prediction'); pnew = pnew';
46% Normalization of all variables (values between −1 y and +1)
47pnewn = tramnmx(pnew,minp,maxp);
48% Testing the ANN obtaining the properties for the variables provided
49anewn = sim(net,pnewn);
50% Transformation of the normalized exits (between −1 y and +1) determined by the ANN to real values
51anew = postmnmx(anewn,mint,maxt); anew = anew';
52% Saving the predicted properties in an Excel file
53dlmwrite('solub_ predicted.xls',anew,char(9));
```

To avoid overfitting, a limit of up to 10 neurons per hidden layer is considered. The number of epochs for training the algorithms is set to 900. For each architecture,

50 executions are run. In addition, all selected architectures were evaluated on the testing and prediction datasets to ensure their performance.

In this study, for the selection of the learning algorithm, eight different functions are employed: the BFGS quasi-Newton, resilient backpropagation, scaled conjugate gradient, conjugate gradient with P/B, Polak–Ribière conjugate gradient, one-step secant, variable learning rate backpropagation, and Levenberg–Marquardt. The criteria used are performance, convergence, and statistical parameters. In this stage, a simple architecture of (4,2,2,1) is used. The best algorithms are as follows: BFGS quasi-Newton, with a performance of 0.027 and average absolute deviations of 17.40% for training and 18.24% for testing; Levenberg–Marquardt, with a performance of 0.007 and average absolute deviations of 17.88% for training and 18.77% for testing; and Polak–Ribière conjugate gradient, with a performance of 0.0031 and average absolute deviations of 18.78% for training and 19.55% for testing. In Figure 1, we can see that the BFGS quasi-Newton, Levenberg–Marquardt, and Polak–Ribière conjugate gradient functions have acceptable convergence speeds. Table 4 shows the results obtained by all the algorithms used in this work.

**Table 4.** Best performance and results in training and testing with different algorithms used in this work.

| Algorithm | Training Function | Run | Best Performance | $\lvert \Delta x_1 \% \rvert_{\text{Training}}$ | $\lvert \Delta x_1 \% \rvert_{\text{Testing}}$ |
|---|---|---|---|---|---|
| Levenberg–Marquardt | trainlm | 14 | 0.007 | 17.88 | 18.77 |
| BFGS quasi-Newton | trainbfg | 37 | 0.027 | 17.40 | 18.24 |
| One-step secant | trainoss | 10 | 0.0038 | 19.16 | 22.29 |
| Resilient backpropagation | trainrp | 13 | 0.0042 | 18.46 | 20.05 |
| Scaled conjugate gradient | trainscg | 16 | 0.0031 | 18.22 | 19.32 |
| Fletch–Powell conjugate gradient | traincgf | 5 | 0.0032 | 19.16 | 20.17 |
| Polak–Ribière conjugate gradient | traincgp | 26 | 0.0031 | 18.78 | 19.55 |
| Variable learning rate | traingdx | 21 | 0.0076 | 23.44 | 22.76 |

For the optimization process of the hidden layers, architectures of the form (4,m,n,1) are considered. The 1rs hidden layer is optimized separately, using the three previously selected algorithms: BFGS quasi-Newton, Levenberg–Marquardt, and Polak–Ribière conjugate gradient. In addition, a fixed value of n = 2 is considered, and (4,m,2,1)-type architectures are studied. In Figure 2, the results obtained for the average absolute deviation in training for each m are shown. The figure shows that the Levenberg–Marquardt algorithm performs well for all the architectures studied. In comparison with the performance of the BFGS quasi-Newton and scaled conjugate gradient algorithms, the Levenberg–Marquardt algorithm in the architectures with m < 7 gives a considerably better performance. However, the results of the BFGS quasi-Newton algorithm, with m = 7, 8, and 10, match the results of the Levenberg–Marquardt algorithm. The findings show that, for m = 4, 5, and 6, an acceptable average absolute deviation is achieved in training with the Levenberg–Marquardt algorithm (5.68%, 4.61%, and 4.33% in the training dataset, respectively). Therefore, using the criterion of selecting architectures with a reduced number of parameters, the Levenberg–Marquardt function is selected for the (4,4,n,1), (4,5,n,1), and (4,6,n,1) architectures.

For the optimization of the second hidden layer, we study combinations of neurons in the hidden layers, considering m = 4, 5, and 6; and n = 2, 3, . . ., 10. Table 5 presents the run with the lowest average absolute deviation during training, testing, and prediction for each architecture. A considerable number of architectures achieve an average absolute deviation of less than 5% (85% of architectures in training, 74% of architectures in testing, and 96% of architectures in prediction).

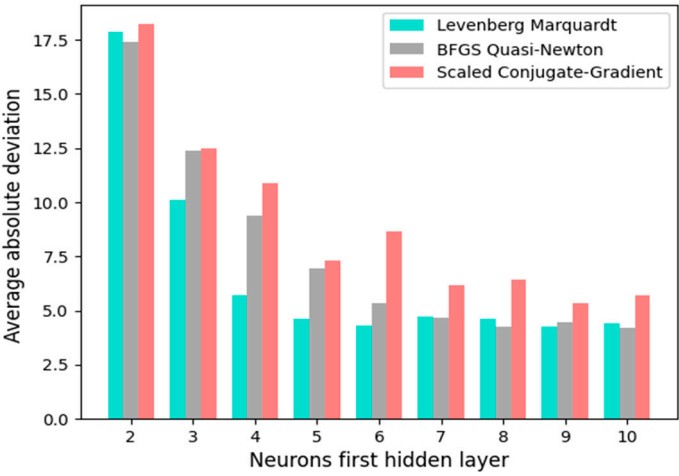

**Figure 2.** Optimization of the first hidden layer for architectures of type (4,m,2,1) with m = 1, 2, 3 . . ., 10.

**Table 5.** Results of the (4,6,n,1), (4,5,n,1), and (4,4,n,1) architectures using T, P, Tc, and Pc training variables (Np: number of parameters).

| Architecture | Np | Run | Training (396 Data Point) | | Testing (22 Data Point) | | Prediction (22 Data Point) | |
|---|---|---|---|---|---|---|---|---|
| | | | $|\Delta x_1\%|$ | $|\Delta x_1\%|_{max}$ | $|\Delta x_1\%|$ | $|\Delta x_1\%|_{max}$ | $|\Delta x_1\%|$ | $|\Delta x_1\%|_{max}$ |
| 4,6,2,1 | 47 | 50 | 4.33 | 57.43 | 3.49 | 14.87 | 3.78 | 16.81 |
| 4,6,3,1 | 55 | 1 | 4.83 | 59 | 3.89 | 23.16 | 4.37 | 16.79 |
| 4,6,4,1 | 63 | 21 | 4.2 | 41.23 | 4.48 | 19.57 | 3.55 | 15 |
| 4,6,5,1 | 71 | 5 | 3.31 | 30.33 | 4.16 | 26.33 | 2.77 | 13.28 |
| 4,6,6,1 | 79 | 27 | 4.76 | 56.52 | 3.64 | 13.14 | 3.86 | 18.59 |
| 4,6,7,1 | 87 | 13 | 4.28 | 63.36 | 4.85 | 13.08 | 3.1 | 9.8 |
| 4,6,8,1 | 95 | 6 | 4.5 | 54.71 | 5 | 14.51 | 3.78 | 15.68 |
| 4,6,9,1 | 103 | 48 | 4.61 | 47.92 | 4.42 | 13.69 | 3.22 | 15.16 |
| 4,6,10,1 | 111 | 26 | 3.63 | 37.64 | 5.78 | 20 | 1.99 | 10.17 |
| 4,5,2,1 | 40 | 42 | 4.61 | 32.29 | 5.86 | 29.86 | 3.58 | 15.76 |
| 4,5,3,1 | 47 | 33 | 5.06 | 37.2 | 4.76 | 22.9 | 3.87 | 14.89 |
| 4,5,4,1 | 54 | 28 | 3.81 | 27.07 | 3.81 | 17.31 | 3.68 | 13.33 |
| 4,5,5,1 | 61 | 31 | 5.17 | 49.05 | 4.62 | 16.53 | 3.38 | 15.18 |
| 4,5,6,1 | 68 | 9 | 4.32 | 51.52 | 4.71 | 35.03 | 3.34 | 10.67 |
| 4,5,7,1 | 75 | 44 | 4.44 | 37.39 | 4.9 | 18.89 | 3.54 | 12.82 |
| 4,5,8,2 | 82 | 1 | 4.56 | 30.84 | 3.41 | 11.93 | 3.49 | 12.95 |
| 4,5,9,2 | 89 | 14 | 4.25 | 36.35 | 5.86 | 27.55 | 2.93 | 9.97 |
| 4,5,10,2 | 96 | 18 | 3.44 | 33.84 | 4.44 | 12.74 | 2.24 | 7.19 |
| 4,4,2,1 | 33 | 5 | 5.68 | 55.22 | 6.2 | 23.08 | 5.86 | 21.88 |
| 4,4,3,1 | 39 | 38 | 7.52 | 57.43 | 6.92 | 40.37 | 4.63 | 16.96 |
| 4,4,4,1 | 45 | 15 | 3.59 | 25.12 | 3.57 | 13.12 | 2.2 | 11.92 |
| 4,4,5,1 | 51 | 33 | 4.26 | 33.38 | 4.62 | 23.7 | 3.42 | 13.36 |
| 4,4,6,1 | 57 | 21 | 3.82 | 34.47 | 3.96 | 25.12 | 3.22 | 11.01 |
| 4,4,7,1 | 63 | 3 | 3.4 | 31.75 | 5.02 | 25 | 2.48 | 9.71 |
| 4,4,8,1 | 69 | 31 | 4.64 | 41.64 | 4.65 | 18.91 | 3.04 | 17.44 |
| 4,4,9,1 | 75 | 48 | 3.62 | 23.49 | 4.12 | 17.49 | 3.09 | 10.51 |
| 4,4,10,1 | 81 | 28 | 2.87 | 24.72 | 3.44 | 17.71 | 2.49 | 11.79 |

In general, the results show that, for n = 10, low average absolute deviations are obtained. However, for the case of architectures of type (4,4,n,1), the lowest average absolute deviation, with a prediction value of 2.2%, is reached for n = 4, which also presents acceptable values in training and testing (3.59% and 3.57%, respectively). On the other hand, for architectures of type (4,5,n,1), the lowest average absolute deviations in training and

prediction are obtained for n = 10 (3.44% and 2.24%, respectively). Additionally, acceptable results are obtained for architectures with fewer parameters, such as architectures with n = 4 and 8. Finally, for architectures of type (4,6,n,1), for n = 10, a value of less than 2.0% in the average absolute deviation in prediction is achieved. However, other simpler architectures show a better performance in regard to training and testing. In particular, for the architecture with n = 5, we obtain 3.31% and 4.16% in training and testing, respectively.

The results show that, by increasing the number of neurons in the hidden layers and, thus, the number of network parameters, the average absolute deviations are not significantly reduced. In particular, simple architectures such as (4,4,4,1), (4,5,4,1), and (4,6,5,1) employ a low number of parameters (45, 54, and 71 parameters, respectively) and obtain acceptable results. Therefore, based on the above criteria, the (4,4,4,1) architecture is selected as the best model. Figure 3 shows the correlation between the experimental and calculated solubilities for architecture (4,4,4,1).

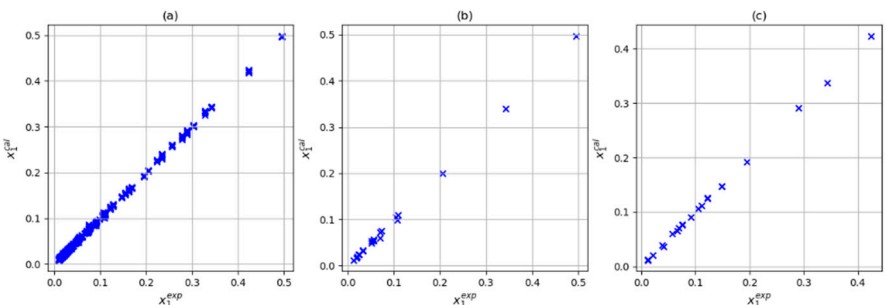

**Figure 3.** Correlation between experimental data and calculated by ANN, using T, P, Tc, and Pc inputs with (4,4,4,1) architecture: (**a**) training dataset, (**b**) testing dataset, and (**c**) prediction dataset.

To complement this study, three other combinations of inputs are considered: T-P-Tc-Pc-$\omega$, T-P-Tc-Pc-Zc, and T-P-Tc-Pc-$\omega$-Zc. When we add the acentric factor, $\omega$, as a training variable (49 parameters), an increase in the average absolute deviations in all datasets (4.46%, 5.25%, and 3.21% in training, testing, and prediction, respectively) is observed with the (4,4,4,1) architecture. By adding the critical compressibility factor, Zc (49 parameters), an increase in the average absolute deviations in all datasets (4.77%, 6.03%, and 3.72% in training, testing, and prediction, respectively) is observed. Finally, when we add $\omega$ and Zc (59 parameters), we do not observe a better performance of the neural network (4.01%, 4.69%, and 2.89% in training, testing, and prediction, respectively). In Figures 4–6, the correlation between the experimental and calculated solubilities for the T-P-Tc-Pc-$\omega$, T-P-Tc-Pc-Zc, and T-P-Tc-Pc-$\omega$-Zc combinations is presented. In addition, Figure 7 shows that the largest relative deviation is found in the low-solubility region in four cases in the training set in four input combinations. This is reasonable due to the experimental uncertainty inherent in these measurements.

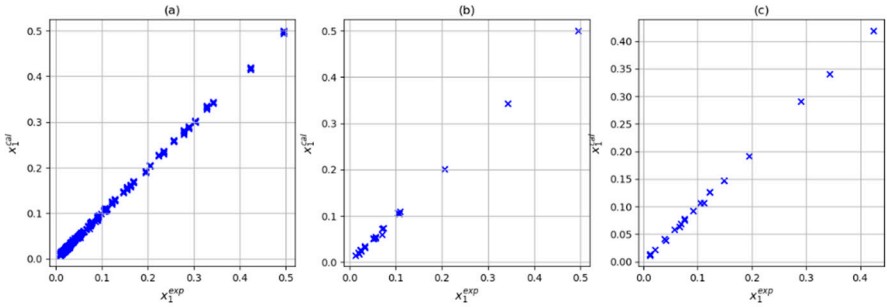

**Figure 4.** Correlation between experimental data and calculated by ANN, using T, P, Tc, Pc, and $\omega$ inputs with (4,4,4,1) architecture: (**a**) training dataset, (**b**) testing dataset, and (**c**) prediction dataset.

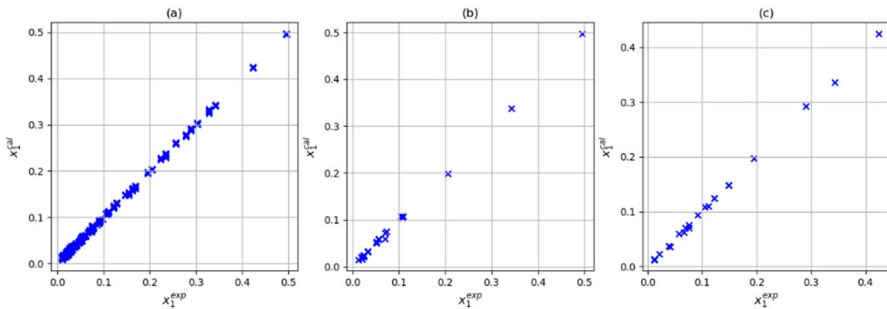

**Figure 5.** Correlation between experimental data and calculated by ANN, using T, P, Tc, Pc, and Zc inputs with (4,4,4,1) architecture: (**a**) training dataset, (**b**) testing dataset, and (**c**) prediction dataset.

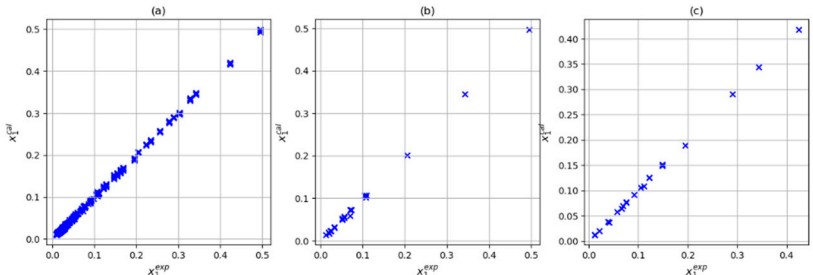

**Figure 6.** Correlation between experimental data and calculated by ANN, using T, P, Tc, Pc, $\omega$, and Zc inputs with (5,4,4,1) architecture: (**a**) training dataset, (**b**) testing dataset, and (**c**) prediction dataset.

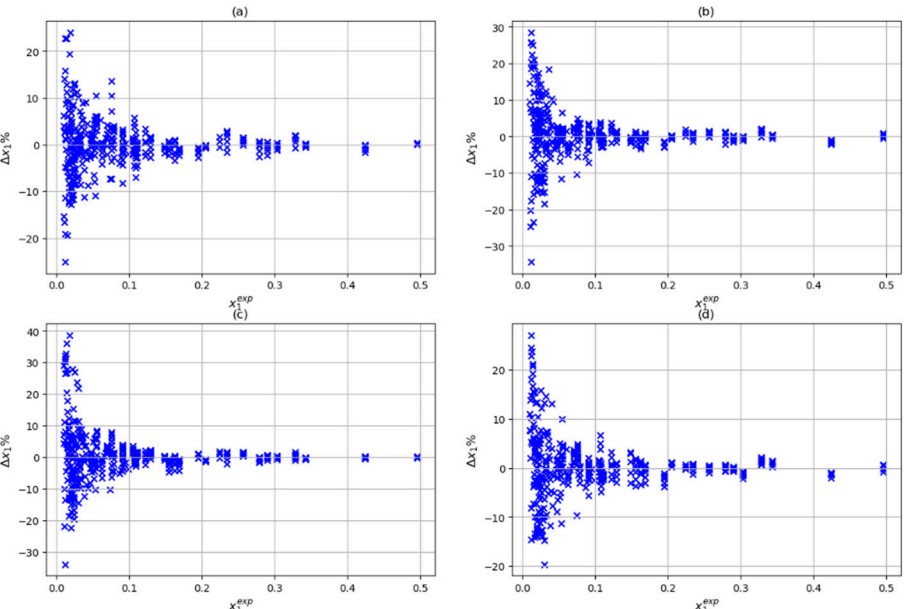

**Figure 7.** Dispersion of training in four input combination studies with (4,4,4,1) architecture: (**a**) T, P, Tc, and Pc; (**b**) T, P, Tc, Pc, and $\omega$; (**c**) T, P, Tc, Pc, and Zc; and (**d**) T, P, Tc, Pc, $\omega$, and Zc.

With these results, we can consider the variables T, P, Tc, and Pc to be the best choice of input combination with the best training, testing, and prediction results. If the critical properties (Tc and Pc) are known for an IL not considered in the training process, it is possible to incorporate them to predict the solubility of new systems with the model proposed in this work [7,24,25]. Moreover, it is a simple model with 45 parameters that predicts solubility with absolute deviations less than 12%.

Conventional methods for modeling the experimental solubility data of $CH_4$ + LI systems are presented in the literature. Althuluth et al. (2017) modeled the solubility of

methane in [C6mim][TCM], using the PR-EoS, with only one temperature-independent binary interaction parameter obtaining an average absolute deviation of less than 2% [48]. Alcantara et al. (2018) reported the correlation of systems composed of methane and [BHEA][Bu], using the three-parametric Redlich–Kwong/Peng–Robinson [49]. The model adjustment resulted in average deviations from data below 10% for molar fraction. Unfortunately, these studies report the overall results as the average absolute deviation, making it difficult to generate a direct comparison. However, the results obtained in this study with artificial neural networks are acceptable in terms of the average absolute deviations. Table 6 shows the results presented by other authors for the modeling of methane solubility in ionic liquids.

**Table 6.** Results reported in the literature for modeling experimental data for mixtures composed of $CH_4$ and ionic liquids.

| Ionic liquid | $T_{range}$ (K) | $P_{range}$ (Mpa) | Model | Comments | Ref. |
|---|---|---|---|---|---|
| [C6mim][TCM] | 293–363 | Up to 10 | Peng–Robinson EoS with only one temperature-independent binary interaction parameter. | The calculated results are in a good agreement with the experimental data, with an average absolute deviation of less than 2%. | [48] |
| [m2HEA][Pr] | 331–363 | 4–16 | Redlich–Kwong/Peng–Robinson EoS coupled to cubic van der Waals mixing rules. | The average error for the mole fraction of methane was around 9.7%. | [49] |
| [m-2HEA][Pr] [BHEA][Bu] | 313.1–353.1 | Up to 20 | Redlich–Kwong/Peng–Robinson equation of state (RKPR-EoS). | The model adjustment resulted in average deviations from data below 10% for molar fraction. | [50] |
| [C6mim][NO3] | 293.1–342.15 | Up to 4 | Extended Henry's law model. | Data were correlated with a reasonable accuracy. The average absolute relative deviation in fugacity was 0.257%. | [41] |
| [C2mim][dep] [thtdp][phos] [thtdp][dca] [amim][dca] [bmpyrr][dca] [cprop][dca] [cprop][Tf₂N] [bmpip][Tf₂N] [tes][Tf₂N] [toa][Tf₂N] | 303.15–363.15 | Up 14 | Peng–Robinson equation of state in combination with van der Waals mixing rules. | They compared the experimental results with those of the model by means of graphical representations. However, they did not present the deviations obtained for these systems. | [40] |
| [C2mim][EtSO4 | 293 K | 0.2–10 | Group contribution equation of state. | Average deviation between experimental and calculated equilibrium pressures of 2.3%. | [51] |

## 4. Conclusions

In this work, the experimental solubility data of 14 binary mixtures composed of $CH_4$ and different ionic liquids were used to train a multilayer perceptron model. The temperatures ranged from 293.15 K to 449.12 K, the pressure ranged from 0.400 MPa to 16.105 MPa, and the solubility ranged from 0.001 to 0.496. The study included 440 experimental data points (P–T-x data). To determine the parameters of the artificial neural network model, eight learning algorithms were studied, and (l,m,n,1)-type architectures were tested. This allowed for the following main conclusions to be drawn. (1) The resulting statistical values indicate that the Levenberg–Marquardt algorithm provided a more accurate nonlinear predictive model. (2) For m = 4 and n = 4, an acceptable mean absolute deviation was

achieved (3.59% in the training dataset, 3.57% in the test dataset, and 2.20% in the prediction dataset). (3) The combination of T, P, Tc, and Pc was a reasonable choice of input with 45 parameters.

**Supplementary Materials:** The following supporting information can be downloaded at: https://www.mdpi.com/article/10.3390/pr12030539/s1, Database used in this work and calculated weights and bias matrices.

**Author Contributions:** Conception and design of study, E.N.F. and C.A.F.; acquisition of data, A.S.M. and E.N.F.; analysis and interpretation of data, E.N.F., C.A.F. and A.S.M.; drafting the manuscript, E.N.F., C.A.F. and A.S.M.; revising the manuscript critically for important intellectual content, E.N.F., C.A.F. and A.S.M. All authors have read and agreed to the published version of the manuscript.

**Funding:** This research was funded by ANID grant number 21171075, research grant DIUA 238-2022 of the VRID, and research grant VRID 19011062-INV.

**Data Availability Statement:** Data are contained within the article and Supplementary Materials.

**Acknowledgments:** The authors are grateful for the support of their respective institutions and grants. E.N.F. gives thanks for the support of ANID scholarship 21171075. C.A.F. and A.S.M. thanks for the support through the research grant VRID N° 219.011.062-INV. The authors acknowledge the DIUA 238-2022 project of the VRID for supporting part of this work. A.S.M. thanks the research group GEMA Res.180/2019 VRID-UA for special support.

**Conflicts of Interest:** The authors declare no conflict of interest.

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
