# Peer review of "Solubility of Methane in Ionic Liquids for Gas Removal Processes Using a Single Multilayer Perceptron Model"

_processes, doi:10.3390/pr12030539_

Round 1

Reviewer 1 Report

Comments and Suggestions for Authors

Thesis entitled Solubility of methane in ionic liquids for gas removal processes using a single multilayer perceptron model. 

It presents new methods for the description of ionic liquids and their application in gas removal. The work contains many new descriptions allowing the use and description of ionic liquids in the described method. 

The authors presented and described the use of ionic liquids for gas removal in a very detailed manner. 

The work provides many interesting mathematical descriptions not previously seen in the literature to the process in question. 

The authors followed the literature references very carefully and used them very well in their work. 

The authors presented mathematical equations for the analysis of the discussed processes in a very clear way. 

The conclusions included in the work precisely reflect the discussed processes presented in the work. 

The work brings many innovative aspects to the topic of ionic liquids and will certainly have a wide range of applications. 

However, the authors made several mistakes. 

There are many editorial errors in the work related to subscripts. they appear in relationship patterns. 

Charts 1 and 2 should be improved. In chart 1, please remove the grid lines, and in chart 2, which is not very clear, please change the colors. 

In my opinion, the authors should also significantly reduce the part of the article devoted to the introduction. 

After applying these corrections, the article should be published.

Author Response

Thank you for your comments. Each question is answered below:

  1. There are many editorial errors in the work related to subscripts. they appear in relationship patterns. 

Answer: The subscripts were reviewed and corrected.

  1. Charts 1 and 2 should be improved. In chart 1, please remove the grid lines, and in chart 2, which is not very clear, please change the colors. 

Answer: In Figure 1 the grid lines were removed and in Figure 2 the colors were changed as indicated by the referee.

  1. In my opinion, the authors should also significantly reduce the part of the article devoted to the introduction. 

Answer: The paragraph on vapor-liquid equilibrium studies of mixtures composed of methane and ionic liquids was reduced in the introduction section.

Reviewer 2 Report

Comments and Suggestions for Authors

The manuscript may be accept after minor revision given below,

The author must use one type of abbreviation, either emim or C2min. Some IL are in the C6py and some emim, confusing authors.

The abbreviation should be defined in the first use, and use only abbreviations. Ionic liquid may be abbreviated as IL and later IL.

Why did the author choose the temperature ranges of 293.15 K and 449.12 K. It must be clarified. And also the pressure range

Author Response

Thank you for your comments. Each question is answered below:

  1. The author must use one type of abbreviation, either emim or C2min. Some IL are in the C6py and some emim, confusing authors.

Answer: In the current version of the manuscript, the IUPAC name of the ionic liquid is incorporated in Table 3 and the specific nomenclature indicating the number of carbons (C2mim, C6mim, etc.) is used.

Table 3. Critical properties acentric factors and compressibility factors of all the substances used in this study (Valderrama et. al (2015)). 
System 
IUPAC name
Tc 
Pc 
Zc 
ω 
[C4mim][Tf2N] 
1-Butyl-3-methylimidazolium bis(trifluoromethylsulfonyl)imide
1258.9 
27.64 
0.2592 
0.3370 
[C4py][BF4
1-Butylpyridinium tetrafluoroborato
597.6 
20.33 
0.2652 
0.8207 
[C4py][Tf2N] 
1-Butylpyridinium bis(trifluorometanosulfonyl)imide
1229.1 
27.71 
0.2666 
0.2505 
[C6py][Tf2N] 
1-Hexylpyridinium bis(trifluorometanosulfonyl)imide
1252.3 
23.93 
0.2522 
0.3383 
[C2mim][dep] 
1-ethyl-3-methylimidazolium diethylphosphate
877.2 
21.47 
0.2349 
0.7219 
[C2mim][FAP] 
1-ethyl-3-methylimidazolium tris(perfluoroethyl)trifluorophosphate
740.6 
10.05 
0.1944 
1.3993 
[C6mim][NO3
1-Hexyl-3-methylimidazolium nitrate
991.8 
23.16 
0.2135 
0.7242 
[C6mim][Tf2N] 
1-Hexyl-3-methylimidazolium bis(trifluoromethylsulfonyl)imide
1293.3 
23.89 
0.2454 
0.3874 
[hmpy][Tf2N] 
1-Hexyl-1-methylpyrrolidinium bis(trifluorometanosulfonyl)imide
1265.2 
22.25 
0.2439 
0.4060 
[tes][Tf2N] 
triethylsulfonium bis(trifluoromethylsulfonyl)imide
1189.9 
21.90 
0.2317 
0.1603 
[thtdp][dca] 
trihexyltetradecylphosphonium dicyanamide
1505.8 
7.65 
0.1388 
1.0319 
[thtdp][phos] 
trihexyltetradecylphosphonium bis(2,4,4-trimethylpentyl)phosphinate
1819.5 
5.51 
0.1157 
0.0924 
[toa][Tf2N] 
methyltrioctylammonium bis(trifluoromethylsulfonyl)imide
1347.6 
10.64 
0.1988 
1.6063 
TMGL 
1, 1, 3, 3-tetramethylguanidium lactate
816.9 
27.18 
0.2557 
1.1188 
  1. The abbreviation should be defined in the first use, and use only abbreviations. Ionic liquid may be abbreviated as IL and later IL.

Answer: The new version of the article defines the relevant concepts and then uses only the abbreviations as indicated by the referee.

  1. Why did the author choose the temperature ranges of 293.15 K and 449.12 K. It must be clarified. And also the pressure range.

Answer: The pressure and temperature ranges used in this work were selected to expand the predictive capacity of the artificial neural network. It is known in the literature that neural networks are not known for extrapolating data, so it is necessary to expand the range of temperatures and pressures considered in the training process. This facilitates the prediction of new data within the range of temperatures and pressures. This was incorporated into the new version of the manuscript on lines 122-125:

“To ensure efficient training of the neural network without sacrificing the model's generalization capability, the temperature and pressure range considered in this work is 293.15-449.30 K and 0.400-16.105 MPa, respectively.”

Reviewer 3 Report

Comments and Suggestions for Authors

Fierro's group presented a machine learning research outcome for evaluating methane solubility in different types of ionic liquid. The concept/approach presented in this article is somewhat similar to a previously reported article by the same group within the same journal (Processes 2022, 10(9), 1686), targeting carbon dioxide.

Some comments for the authors:

Technical aspect of the article - the article contains plenty of missing chemical names and their respective abbreviations, which caused difficulties to readers to follow the types of ionic liquids used. If this article is aimed to cover readers from various disciplines, proper abbreviation system should be present in the article. Also, some inconsistencies in the naming were found, authors to check.

Experimental aspect of the article - The approach is good that it has 14 systems involved in the study. However, the selection of these 14 systems were not clearly highlighted/summarized. Additionally, the hypotheses were not clear. The selection of the ionic liquids were not systematic as well, why there was no trend involved as part of the selection? The overall outcome of the investigation was not clear as well. Was the anion or the cation playing a more significant role in the solubility performance? Does the size of the cations or anions affect the solubility performance? These conclusions are important and should be part of the data analysis.

Comments on the Quality of English Language

Suggestion to authors - proofreading will be required.

Author Response

Thank you for your comments. Each question is answered below:

  1. Technical aspect of the article - the article contains plenty of missing chemical names and their respective abbreviations, which caused difficulties to readers to follow the types of ionic liquids used. If this article is aimed to cover readers from various disciplines, proper abbreviation system should be present in the article. Also, some inconsistencies in the naming were found, authors to check.

Answer: Table 3 includes the IUPAC names of the ionic liquids considered in this work. Additionally, the nomenclature of the ionic liquids is updated indicating the number of carbons (C2mim, C6mim, etc) to avoid confusion.

Table 3. Critical properties acentric factors and compressibility factors of all the substances used in this study (Valderrama et. al (2015)). 
System 
IUPAC name
Tc 
Pc 
Zc 
ω 
[C4mim][Tf2N] 
1-Butyl-3-methylimidazolium bis(trifluoromethylsulfonyl)imide
1258.9 
27.64 
0.2592 
0.3370 
[C4py][BF4
1-Butylpyridinium tetrafluoroborato
597.6 
20.33 
0.2652 
0.8207 
[C4py][Tf2N] 
1-Butylpyridinium bis(trifluorometanosulfonyl)imide
1229.1 
27.71 
0.2666 
0.2505 
[C6py][Tf2N] 
1-Hexylpyridinium bis(trifluorometanosulfonyl)imide
1252.3 
23.93 
0.2522 
0.3383 
[C2mim][dep] 
1-ethyl-3-methylimidazolium diethylphosphate
877.2 
21.47 
0.2349 
0.7219 
[C2mim][FAP] 
1-ethyl-3-methylimidazolium tris(perfluoroethyl)trifluorophosphate
740.6 
10.05 
0.1944 
1.3993 
[C6mim][NO3
1-Hexyl-3-methylimidazolium nitrate
991.8 
23.16 
0.2135 
0.7242 
[C6mim][Tf2N] 
1-Hexyl-3-methylimidazolium bis(trifluoromethylsulfonyl)imide
1293.3 
23.89 
0.2454 
0.3874 
[hmpy][Tf2N] 
1-Hexyl-1-methylpyrrolidinium bis(trifluorometanosulfonyl)imide
1265.2 
22.25 
0.2439 
0.4060 
[tes][Tf2N] 
triethylsulfonium bis(trifluoromethylsulfonyl)imide
1189.9 
21.90 
0.2317 
0.1603 
[thtdp][dca] 
trihexyltetradecylphosphonium dicyanamide
1505.8 
7.65 
0.1388 
1.0319 
[thtdp][phos] 
trihexyltetradecylphosphonium bis(2,4,4-trimethylpentyl)phosphinate
1819.5 
5.51 
0.1157 
0.0924 
[toa][Tf2N] 
methyltrioctylammonium bis(trifluoromethylsulfonyl)imide
1347.6 
10.64 
0.1988 
1.6063 
TMGL 
1, 1, 3, 3-tetramethylguanidium lactate
816.9 
27.18 
0.2557 
1.1188 
  1. Experimental aspect of the article - The approach is good that it has 14 systems involved in the study. However, the selection of these 14 systems were not clearly highlighted/summarized. Additionally, the hypotheses were not clear. The selection of the ionic liquids were not systematic as well, why there was no trend involved as part of the selection?

As a selection criterion, those systems whose critical properties have been calculated with the method proposed by Valderrama (2015) were considered. In the current version of the manuscript, line 126-127 of the introduction clarifies this point,

“In this study, only systems whose critical properties have been reported by Valderrama et al. (2015) were selected [36]”

  1. The overall outcome of the investigation was not clear as well. Was the anion or the cation playing a more significant role in the solubility performance? Does the size of the cations or anions affect the solubility performance? These conclusions are important and should be part of the data analysis.

Answer: The study suggested by the referee could be carried out, however it deviates from the objective of the present work. The interest of the work is to use as input variables to estimate the solubility in the binary mixture CH4 +LI, only equilibrium properties known and available in the literature.
